# Farnesoid X Receptor, Bile Acid Metabolism, and Gut Microbiota

**DOI:** 10.3390/metabo12070647

**Published:** 2022-07-14

**Authors:** Hideki Mori, Gianluca Svegliati Baroni, Marco Marzioni, Francesca Di Nicola, Pierangelo Santori, Luca Maroni, Ludovico Abenavoli, Emidio Scarpellini

**Affiliations:** 1T.A.R.G.I.D., Gasthuisberg University Hospital, KU Leuven, Herestraat 49, 3000 Leuven, Belgium; hideki.mori@kuleuven.be; 2Hepatic Damage and Transplant Unit, Università Politecnica delle Marche, 60121 Ancona, Italy; gsvegliati@gmail.com; 3Gastroenterology Clinic; Università Politecnica delle Marche, 60121 Ancona, Italy; m.marzioni@univpm.it (M.M.); luca.maroni@ospedaliriuniti.marche.it (L.M.); 4Hepatology Outpatient Clinic and Internal Medicine Unit, “Madonna del Soccorso” General Hospital, 63074 San Benedetto del Tronto, Italy; francesca.dinicola@yahoo.com (F.D.N.); pierangelo.santori@sanita.marche.it (P.S.); 5Department of Health Sciences, University “Magna Græcia”, 88100 Catanzaro, Italy; l.abenavoli@unicz.it

**Keywords:** farnesoid X receptor, bile acids metabolism, gut microbiota, dysbiosis, liver steatosis

## Abstract

Obesity, type 2 diabetes, and non-alcoholic fatty liver disease (NAFLD) are characterized by the concepts of lipo- and glucotoxicity. NAFLD is characterized by the accumulation of different lipidic species within the hepatocytes. Bile acids (BA), derived from cholesterol, and conjugated and stored in the gallbladder, help the absorption/processing of lipids, and modulate host inflammatory responses and gut microbiota (GM) composition. The latter is the new “actor” that links the GI tract and liver in NAFLD pathogenesis. In fact, the discovery and mechanistic characterization of hepatic and intestinal farnesoid X receptor (FXR) shed new light on the gut–liver axis. We conducted a search on the main medical databases for original articles, reviews, meta-analyses of randomized clinical trials, and case series using the following keywords, their acronyms, and their associations: farnesoid X receptor, bile acids metabolism, gut microbiota, dysbiosis, and liver steatosis. Findings on the synthesis, metabolism, and conjugation processes of BAs, and their action on FXR, change the understanding of NAFLD physiopathology. In detail, BAs act as ligands to several FXRs with GM modulation. On the other hand, the BAs pool is modulated by GM, thus, regulating FXRs functioning in the frame of liver fat deposition and fibrosis development. In conclusion, BAs passed from their role of simple lipid absorption and metabolism agents to messengers between the gut and liver, modulated by GM.

## 1. Introduction

Obesity, type 2 diabetes, and dyslipidemia are features of our Westernized lifestyle. Non-alcoholic fatty liver disease (NAFLD) follows the same epidemiological trends as these disorders, and has a global prevalence of 25% among the general population [1]. NAFLD is not a status, but rather a stage of conditions including liver steatosis and non-alcoholic steatohepatitis (NASH), all the way up to cirrhosis and hepatocellular carcinoma [1]. Liver fibrosis is a key factor for liver-related, and all-cause, mortality [2].

NAFLD is characterized by the accumulation of different lipidic species within the hepatocytes—a process regarded as “lipotoxicity” [3,4]. In addition, carbohydrates boost several lipogenic pathways (e.g., acetyl-CoA carboxylase, SCD-1, and fatty acid synthase) and contribute to liver steatosis [5]. Fructose is the most effective example of this detrimental interaction with lipotoxicity, and outlines the “glucotoxicity” concept. In fact, these changes are associated with reduced insulin sensitivity, strictly linked to NASH origin [6,7]. 

The gut is linked to the liver through the gut–liver axis [4], and gut microbiota is the link connecting these two organs. Indeed, fat accumulation in hepatocytes is accompanied by changes in the population of gut microbiota that modulate the pool of bile acids (BA) secreted by the liver and transformed/metabolized within the gut [8,9]. 

This review of the scientific literature aims to describe the metabolism of bile acids and their “vital” cycle, and their role in lipogenesis and lipotoxicity in the liver and NAFLD, with a special focus on the role of the farnesoid X receptor in the pathogenesis and potential treatment of liver steatosis. 

## 2. Results

### 2.1. Bile Acids and Their Metabolism: The “Entero-Hepatic” Cycle

BAs are derived from cholesterol to form primary BAs (cholic acid, or CA) and chenodeoxycholic acid, or CDCA [10]. Subsequently, primary BAs are conjugated with glycine or taurine, with improved solubility, and stored in the gallbladder, ready for excretion in the gut [7,11].

The conjugation process has several functions: it minimizes the passive absorption of BAs, modulates host inflammatory responses and gut microbiota composition, and maintains the gut “eubiosis”, limiting small intestinal bacterial overgrowth (SIBO) incidence within our small bowel as a feature of “dysbiosis” [7,12]. 

Postprandially, BAs are released from the gallbladder into the small bowel, following the secretion of GI cholecystokinin (CCK). BAs are crucial for the emulsification of dietary fat, and enhance the absorption of lipids, sterols, and vitamins [9,13]. Approximately 95% of BAs are actively reabsorbed in the terminal ileum, via the apical sodium-dependent bile acid transporter (ASBT). In the cytoplasm of the enterocyte, BAs bind to the ileal bile acid-binding protein (IBABP). Then, they are excreted into the portal circulation through the organic anion transporter polypeptide (OATPA/B), localized in the basolateral membrane of enterocytes. Further, BAs travel back into the liver through the sodium taurocholate cotransporting polypeptide (NTCP) transporter [7,8,10]. Once in the liver, free BAs are re-conjugated with taurine or glycine, before secretion into the biliary tract and intestinal lumen [10]. This enterohepatic cycle of BAs occurs almost 10 times per day, and is necessary as hepatocytes have limited capability to produce BAs [10] (Figure 1). 

Ileal BA transport is highly efficient, but a small proportion (1–2%) of BA escapes the enterohepatic circulation, and enters the large intestine [10]. As they transit through the colon, the microbiota perform several enzymatic reactions (e.g., deconjugation, dihydroxylation, and epimerization), and form secondary BAs (namely, deoxycholic acid (DCA) from CA, and ursodeoxycholic acid (UDCA) and lithocholic acid (LCA) from CDCA) [7,14]. 

In detail, BAs deconjugation by gut microbiota is crucial for bile bio-transformation [15]: this small portion of BAs being microbially metabolized makes them more lipophilic, allowing the secondary BAs to be reabsorbed in the large bowel and returned to the liver, via the systemic circulation [7,11] (Figure 1).

### 2.2. Bacterial Bile Salt Hydrolase (BSH) Enzymes as a Sign of Bacterial Modulation of BAs Pool

Bile salt hydrolases belong to the Ntn-hydrolase superfamily of proteins [16]. In particular, microbial BSH cleaves the amide bond between the glycine and taurine portions conjugated to the steroid nucleus of bile salts [3,12]. BSH enzymes are represented across most bacterial phyla. Regarding the commensal gut microbiota, BSH activity is found in Gram-positive bacteria: *Lactobacillus*, *Bifidobacterium* [3,17,18], *Clostridium* spp., and *Enterococcus* [19,20]. However, they are also detected in some commensal Gram-negative strains, such as the *Bacteroides* spp [3,12,21,22]. 

BSH presence is also detected in pathogens. For example, the gastrointestinal *Listeria monocytogenes* has BSH activity, an adaptive quality guaranteeing its gut persistence [12,23,24]. Furthermore, BSH activity is also found in the environment: *Xanthomonas maltophilia* is present in soil [25,26], and thermophilic *Brevibacillus* sp in hot springs [27,28]. 

This adaptive quality has a horizontal transmission amongst gut bacteria, suggestive of a strong evolutionary selection for this activity [12,29]. 

Description of functioning, distribution, and abundance of BSH within the human gut microbiome is possible through a metagenomics approach, enabling researchers to reconstruct entire genomic libraries starting from small genomic sequences [30]. More interestingly, BSH coding sequences are found among two domains of life within the gut: bacteria and archaea. Thus, this wide distribution is suggestive of host-driven selection of BSH activity. In addition, the host seems to affect BSH activity in gut bacteria through a “host species-specific selection” of microbial BSH activities. This selective mechanism can be started by species-specific differences in host bile acid pools [31].

From a functional point of view, we can briefly describe the BSH activity as a protective shield for some bacterial species (specifically, through bile acid deconjugation) to colonize the human gut [12,32,33]. The BSH activity products (namely, glycine or taurine) can be used as a source of energy for some bacterial species [12,20,21,22,23,24,25,26,27]. For example, glycine can be metabolized to ammonia and carbon dioxide, and taurine can be metabolized to produce ammonia, carbon dioxide, and sulfate. All of them are carbon and nitrogen sources [12,19,34,35]. Furthermore, BSH regulates bacterial intracellular pH, perpetuating resistance to bile acids in low pH environments (e.g., the stomach) [12]. Finally, BSH is included as a hallmark of probiotic activity, as it allows the strain to survive gut transit [12,36,37].

### 2.3. Bile Acids and Their Receptors: The Emerging “FXR Case”

The concept of the gut–liver axis helps explain the complex interplay between systemic metabolism, gut hormone release, and the immune response [3]. Over the main actors in this system, the arrows making this axis efficient are operated by BAs. Indeed, the latter are also ligands for receptors that include the nuclear receptor farnesoid X receptor (FXR) and G-protein-coupled bile acid receptor 1 (or TGR5). These receptors regulate host basal metabolism and enterohepatic circulation [38].

For example, the strong antimicrobial properties of BAs, and the maintenance of the eubiosis of gut microbiota, are mediated by a multifaceted mechanism of activation of FXR [3,39]. NAFLD offers a mechanistic example of dysregulation of BAs-FXR-mediated lipid and glucose metabolism [3,13]. Indeed, hepatic FXR activation mediated by BAs can induce the expression of atypical nuclear receptors small heterodimer partner (SHP), which promotes the inhibition of the sterol-regulatory element-binding protein-1c (SREBP-1c) (Figure 2), and ultimately leads to the reduced hepatic synthesis of triglycerides. Moreover, FXR can limit fat accumulation in the liver by promoting fatty acid oxidation after the activation of the hepatic NR peroxisome proliferator-activated receptor alpha (PPAR-α), and through plasma VLDL triglyceride clearance (Figure 2) [3,40,41]. FXR activation in the liver shows unidirectional effects on glucose homeostasis. It leads to the inhibition of gluconeogenesis and glycolysis through reduced expression of PEPCK phosphoenolpyruvate carboxykinase (PEPCK) and glucose-6 phosphatase (G6Pase), with a potential protective role in insulin resistance and type II diabetes (Figure 2). However, other studies fail to confirm these actions [42,43]. 

More interestingly, the activation of FXR localized in the intestinal cells can start a crucial endocrine feedback mechanism [3,44]. In fact, the human fibroblast growth factor 19 (FGF19) is secreted into the bloodstream after ileal FXR activation, and suppresses the synthesis of BAs through the activation of the FGF receptor 4 (FGFR4) at the surface of hepatocytes [18]. Its mouse ortholog is FGF15 [3,18]. Practically, FGF 19 signaling cascade reduces liver steatosis and insulin resistance [45]. 

Consequently, evidence of a FXR-mediated decrease in hepatic lipid accumulation and improved insulin sensitivity is the basis for the use of its agonists in NAFLD treatment. For example, a dose of 25 mg/day of the semisynthetic BA obeticholic acid (OA) in patients with non-cirrhotic, non-alcoholic steatohepatitis was compared to placebo for 72 weeks of treatment [46]. As a primary outcome, OA improves liver histology in 46% of patients vs. 21% of patients in the placebo group; no worsening of fibrosis is observed. However, OA is observed to increase plasmatic lipids concentration, and cause pruritus. FXR agonists already show efficacy against primary cholestatic cholangitis, due to the FXR-stimulated reduction in the bile acid pool in the liver, which, in turn, reduces liver inflammation and fibrosis (Figure 2) [40]. Part of the effect of FXR agonists on NASH may be due to this physiological effect. Finally, other FXR agonists (e.g., GS9674 and LJN452) are under investigation [3,47,48] (Table 1). 

The use of FGF-19 and -15 agonists is discouraged because of their correlation with hepatocyte proliferation, potentially leading to hepatocellular carcinoma (HCC) development in mice. Accordingly, human HCC cells and cirrhotic livers show increased expression of FGF19 [3,49]. 

These data remark the need for a better understanding of the pathways induced by FXR activation, before their clinical use.

BAs can also “ sense “ the G-protein-coupled membrane receptor, TGR5, expressed in the gallbladder, the liver, specifically Kupffer cells and endothelial cells, not in hepatocytes, the intestine, adipose tissue, spleen, and the kidneys. TGR5 regulates the composition of BAs acid pool, and modulates the immune system and metabolism [3,50,51]. Interestingly, administration of cholic acid to high-fat diet (HFD)-fed mice results in body weight reduction and enhanced energy expenditure through a TGR5–cAMP pathway characterized in brown adipose tissue and skeletal muscle [52]. BAs also activate TGR5 in enterocytes with the release of GLP-1 and peptide tyrosine PYY, with reduced food intake and normalized glucose metabolism [53].

In addition, INT-767, an FXR/TGR5 dual agonist, shows lowered lipid accumulation, inflammation, and fibrosis in various in vitro models [54,55,56], while studies on individuals with NASH are still ongoing [55] (Table 1). 

### 2.4. Bile Acids, FXR, and Gut Microbiota: The Completion of the Gut-Liver Axis

The intestinal microbiota represents one of the most complex ecosystems present in nature. It accounts for over 100 trillion microorganisms [56]. *Firmicutes* and *Bacteroidetes* represent the most abundant phyla of the microbiota. Their ratio is a fundamental factor in the host’s health [57]. Among the other phyla present, we find *Fusobacteria*, *Actinobacteria*, *Proteobacteria*, and *Verrucomicrobia* [4]. Gut microbiota also encompasses viruses, protozoa, archaea, and fungi [58]. 

Eubiosis defines the balanced qualitative and quantitative condition of the intestinal microflora, and is essential to preserving the host’s health. On the contrary, its qualitative/quantitative perturbation, namely, dysbiosis, is associated with the development of various diseases such as NAFLD, NASH until HCC [59,60], diabetes type 2, cardiovascular disorders, and, undoubtedly, obesity [61,62] (Figure 2). 

As BAs are metabolized by gut microbiota, changes in their composition influence the pathways mediated by BAs, such as FXR signaling [63]. Thus, we can imagine how BAs pool modifications induced by gut microbiota strongly influence host metabolism. In fact, the accumulation of the murine taurine-conjugated primary bile acids tauro-alfa and beta-muricholic acid antagonizes FXR signaling in the ileum in germ-free mice. On the other hand, the physiologic intestinal microbiota presence counteracts these actions, and re-establishes FXR activation with BAs synthesis [3,64] (Figure 2). 

One cutting edge research, unraveling the circular interaction of gut microbiota and BAs via FXR, was led by Friedman et al. [65], through a phase 1 human study involving 17 healthy volunteers administered with OA (a BA analog and FXR agonist) at 5 mg, 10 mg, or 25 mg per day. Fecal and plasma specimens were collected at baseline (day 0) and on days 17 (end of dosing) and 37 (end of study), then analyzed by metagenomic sequencing. Thereafter, a Uniref90 high stringency genomic analysis was performed, in order to match specific genes to specific bacterial species abundant because of OA. At the same time, male C57BL/6 mice were gavaged daily with water, vehicle, or OA (10 mg/kg) for 2 weeks, and small bowel luminal contents were collected at baseline and on day 14. Mouse fecal pellets were analyzed by 16S tagged sequencing. Culture experiments allowed the researchers to determine the species-specific effects of BAs and OA on bacteria. 

Interestingly, OA suppresses endogenous BA synthesis with a reversible induction of Gram-positive bacteria in the small intestine, mainly derived from diet and the oral microbiota. These strains of bacteria are normally down-regulated by the BAs pool. At a molecular level, OA treatment determines an increased activity of microbial genomic pathways involved in DNA synthesis and amino acid metabolism. Accordingly, mice treated with OA show reduced endogenous BA levels, and an increased proportion of *Firmicutes* in the small bowel, as compared to control mice [66]. Oppositely, BAs reduce the proliferation of these bacteria in minimum inhibitory concentration assays. In *Firmicutes*, there is an increase in the representation of microbial genomic pathways involved in DNA synthesis and amino acid metabolism, such as after OA treatment of healthy subjects. There are currently no data showing how BA composition and concentration affect the gut microbiota in the pathogenesis of various diseases, but it is noteworthy that the manipulation of the gut microbiota from BAs may be considered in the future to develop clinical approaches aimed at disease prevention and treatment.

Furthermore, the modulation of intestinal FXR by BAs unravels some interesting consequences for the development of obesity and liver steatosis. Indeed, experimental models of NAFLD show microbiota from obese subjects lowering the levels of BAs with higher activity of FXR. This results in the synthesis of ceramides that alter lipid metabolism at the liver level, leading to the accumulation of fatty acids [3,67]. Accordingly, FXR activation in L cells decreases proglucagon expression interfering with the glucose-responsive factor carbohydrate-responsive element-binding protein (ChREBP), and GLP-1 secretion through glycolysis inhibition [67]. Conversely, agents able to affect gut microbiota and BA composition (e.g., antibiotics, pre-, probiotics) can increase the amount of the intestinal FXR antagonist tauro-alpha-muricholic acid. Finally, the reduction in intestinal FXR activation results in the improvement of both general metabolic and specific hepatic profiles [3,30,31].

In addition, gut microbiota from obese mice has a promotional activity in favor of diet-induced obesity through FXR. Subsequently, FXR may contribute to increased adiposity through action on microbiota composition [68].

Summarizing all these pieces of evidence, we hypothesize that inhibition of FXR within the gut might protect against the development of a fatty liver. However, this hypothesis is in contrast with the assumption of FXR activation being beneficial during NAFLD [69]. Thus, intestinal FXR antagonists are considered for the treatment of metabolic diseases: the glycine-beta-muricholic acid (namely, a tauro-beta-muricholic acid derivative) is used in experimental models of NAFLD, producing a reduction in obesity, insulin resistance, and liver fibrosis/inflammation grading [70,71,72].

In conclusion, we assume that FXR regulation by BAs and gut microbiota is very complex, as is strongly influenced by the organ in which such receptors are activated. 

## 3. Materials and Methods 

We conducted a search on PubMed and Medline for original articles, reviews, meta-analyses, and case series using the following keywords, their acronyms, and their associations: farnesoid X receptor, bile acids metabolism, gut microbiota, dysbiosis, and liver steatosis. When appropriate, preliminary evidence from abstracts belonging to main national and international gastroenterological meetings (e.g., United European Gastroenterology Week, Digestive Disease Week) was also included. The papers found from the above-mentioned sources were reviewed by two of the authors (E.S. and G.S.B.), according to PRISMA guidelines [73]. The last Medline search was dated 15 May 2022.

In total, we found 249 manuscripts matching our search: 69 were clinical trials/original articles, of these 47 randomized clinical trials (RCT)s; 95 reviews of the scientific literature, 1 of those a systematic review; 3 meta-analyses. We included data from 41 original articles (including in vitro and animal studies), 5 RCTs, 26 reviews of the scientific literature, excluding those not updated, and from 1 meta-analysis.

## 4. Conclusions

This review of the scientific literature conveys a novel concept of BAs beyond the classic concept of simple regulators of lipid absorption. They have a sophisticated mechanism of networking with cells not only in the liver, but also in the intestine. This multi-faceted net of chatting is possible through nuclear receptors such as FXR that have at least a bi-valent role in liver fat deposition and metabolism. Moreover, the gut microbiota is crucial for BAs metabolic transformation within the gut. More interesting, gut microbiota composition is modulated by BAs pool and, vice versa; it affects their forms and composition. Thus, this unconventional connection is crucial for the host’s health and survival, as BAs avoid the emergence of pathogenic microbial strains from our intestinal tract. Several factors affecting gut microbiota (e.g., diet, antibiotics) influence its composition, and bring BAs and their interactions with cells towards a dysmetabolic state (e.g., obesity, type II diabetes, NAFLD, NASH), favoring gluco- and lipotoxicity.

Indeed, BAs pool modulation is as crucial as microbial eubiosis for our survival and health. All these pieces of evidence explain the interest of researchers in nuclear receptors modulation with agonists/antagonists according to different tissue targeted in our fight against an obesity-driven Westernized world. Moreover, pre-, pro-, and post-biotics can be powerful and promising weapons to correct and heal a “fattening” BAs pool, through microbial modulation. 

## Figures and Tables

**Figure 1 metabolites-12-00647-f001:**
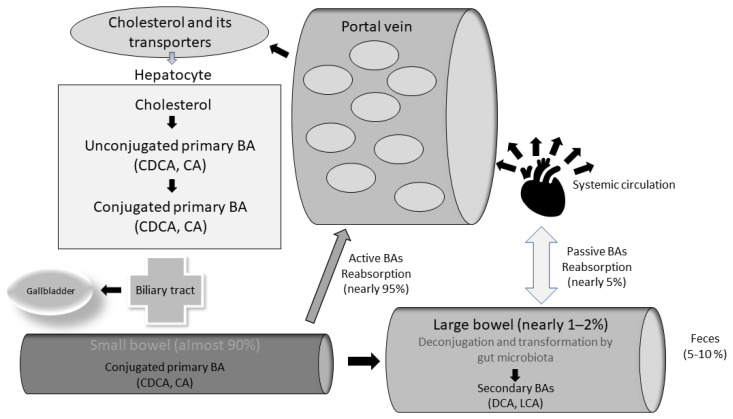
The enterohepatic bile acids cycle.

**Figure 2 metabolites-12-00647-f002:**
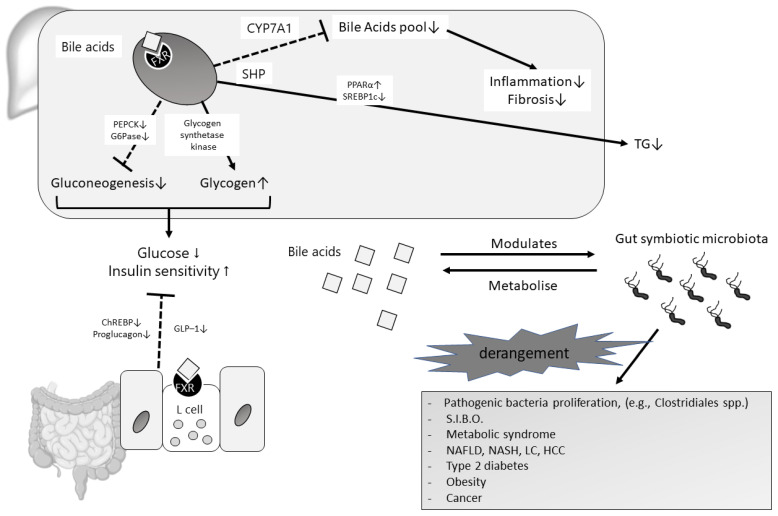
The gut–liver axis representation with special focus on BAs and gut microbiota relationship in healthy and pathologic conditions.

**Table 1 metabolites-12-00647-t001:** Main FXR agonist available and under investigation.

FXR Agonist Product Name	Current Trials Status	Notes on Mechanism of Action	Reference/Protocol Registration
**Obeticholic acid (Ocaliva®)**	FDA/EMA approval for PBC in 2016	*100 times higher affinity than CDCA*	Kowdley KV et al. Hepatology. 2018 [40]
	Efficacy to NASH (phase3)		Neuschwander-Tetri BA et al. Lancet. 2015 [39]
	Bile Acid Malabsorption (phase2)		Walters JR. Aliment Pharmacol Ther. 2015 [48]
**LMB763**	NAFLD (phase2)		NCT02913105
**LJN452**	NAFLD (phase2)		NCT02855164
**GS9674**	NAFLD (phase2)		NCT02854605
	PSC (phase2)		NCT02943460
**WAY-362450**	phase1		NCT00499629. 2007
**PX20606**	phase1		NCT01998659, NCT01998672
**GW4064**	in mice		
**INT-767**	in mice	*dual FXR and TGR5 agonist*	

**Table legend:** PBC: primitive biliary cholangiopathies; NASH: non alcoholic steato-hepatitis; NAFLD: non alcoholic fatty liver disease; PSC: primary sclerosing cholangitis; CDCA: chenodeoxycholic acid; TGR5: G protein-coupled bile acid receptor.

## Data Availability

All the data reported in this review of literature are available online on PubMed and main national and international gastroenterological meetings websites.

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
