# Peer review of "Farnesoid X Receptor, Bile Acid Metabolism, and Gut Microbiota"

_metabolites, 2022, doi:10.3390/metabo12070647_

Round 1
Reviewer 1 Report
The manuscript is a review of literature on franesoid X receptor, bile acid metabolism and gut microbiota. The authors underline the importance of bile acid metabolism, the role played by franesoid X receptor, and how several factors can affect gut microbiota composition and lead to a dysmetabolic state.
The topic is interesting and in line with the journal.
The article is well structured, but a careful reading is recommended to fix some small typos (e.g. correcting the word “al-so” at the line 6 of Materials and Methods section, correcting the word “literuature” at the first line of Conclusions section, removing repeated “: :” at the third line of Materials and Methods section).
Titles and references should be adjusted according to the format of the journal.
In my opinion, the manuscript can be published after minor revision.
Author Response
The manuscript is a review of literature on franesoid X receptor, bile acid metabolism and gut microbiota. The authors underline the importance of bile acid metabolism, the role played by franesoid X receptor, and how several factors can affect gut microbiota composition and lead to a dysmetabolic state.
The topic is interesting and in line with the journal.
The article is well structured, but a careful reading is recommended to fix some small typos (e.g. correcting the word “al-so” at the line 6 of Materials and Methods section, correcting the word “literuature” at the first line of Conclusions section, removing repeated “: :” at the third line of Materials and Methods section).
We thank the reviewer for these comments. We have checked and corrected typos throughout the manuscript.
Titles and references should be adjusted according to the format of the journal.
We thank the reviewer for these comments. We have adjusted title and references according to the journal format.
In my opinion, the manuscript can be published after minor revision.
Reviewer 2 Report
In my opinion, topic of manuscript is suitable for the Metabolites. Nevertheless, the quality of the manuscript needs to be significantly improved.
Role disregulation of bile acid metabolism in the discused pathogenesis should desribed more detailed. For example, their influence on the metabolites production.
Materials and Methods-Results of Pubmed and Medline search (e.g. number of studies-found, excluded, inluded) should be described more detailed.
Figure 2 Efect of bile acid pools on the symbiotic bacterias should also presented in this models. Terms Liver and glucose metabolism, Liver fat deposition, Cell proliferation and Inflammtory response modulation are to generally. Athours should described and discused, not necessarily in this picture, specific effect and its role in the pathogenesis.
Author Response
In my opinion, topic of manuscript is suitable for the Metabolites. Nevertheless, the quality of the manuscript needs to be significantly improved.
Role disregulation of bile acid metabolism in the discused pathogenesis should desribed more detailed. For example, their influence on the metabolites production.
We thank the reviewer for these comments. We have enriched the pathogenesis section of the review.
Materials and Methods-Results of Pubmed and Medline search (e.g. number of studies-found, excluded, inluded) should be described more detailed.
We thank the reviewer for these comments. We have added the required data to the Methods section.
Figure 2 Efect of bile acid pools on the symbiotic bacterias should also presented in this models. Terms Liver and glucose metabolism, Liver fat deposition, Cell proliferation and Inflammtory response modulation are to generally. Athours should described and discused, not necessarily in this picture, specific effect and its role in the pathogenesis.
We thank the reviewer for these comments. We have added the required particulars to Figure 2 and the description of metabolism of BAs within the text.
Reviewer 3 Report
This manuscript written by Mori and co-workers deals with an interesting and timely subject which is the role Farnesoid X receptor in the gut-liver axis.
Major comments:
Abstract: The abstract should better reflect the matter and the flow of the review and gives a clearer overview of the current work.
Proportion of review articles cited: Please cite the original(s) paper(s) instead of review articles when talking about specific results or mechanisms. Among the references cited in this review more than 50% are reviews (or mini-reviews) which is too high. When citing a review, the authors should at least indicate the readers that the information is coming from a review article or instead cite original articles.
With such a methodology described in material and methods, the reader would except more details regarding the papers the authors retrieved and how the studies mentioned in the current review were selected. Hence, it might be worth mentioning some metrics like the number of publications retrieved using the different keywords and their associations to look for them. This might help explaining why the final list of references is made of 62 references including more than 30 reviews.
To wrap up, I think more work could be performed to make this review easier to read/digest.
###########################################################################
Other comments
The manuscript needs more proof-reading, for example
in the title: Franesoid
Page 1: farnsoid
Page 2: al-so
Page 4 : PPAR-alfa
Page 7 : literuature
Author Response
This manuscript written by Mori and co-workers deals with an interesting and timely subject which is the role Farnesoid X receptor in the gut-liver axis.
Major comments:
Abstract: The abstract should better reflect the matter and the flow of the review and gives a clearer overview of the current work.
We thank the reviewer for these comments. We agree with this comment and have changed it accordingly.
Proportion of review articles cited: Please cite the original(s) paper(s) instead of review articles when talking about specific results or mechanisms. Among the references cited in this review more than 50% are reviews (or mini-reviews) which is too high. When citing a review, the authors should at least indicate the readers that the information is coming from a review article or instead cite original articles.
We thank the reviewer for this precious comment. We have modified the references accordingly.
With such a methodology described in material and methods, the reader would except more details regarding the papers the authors retrieved and how the studies mentioned in the current review were selected. Hence, it might be worth mentioning some metrics like the number of publications retrieved using the different keywords and their associations to look for them. This might help explaining why the final list of references is made of 62 references including more than 30 reviews.
We thank the reviewer for these comments. We have added the required data to the Methods section. Furthermore, we have updated the references replacing reviews with original studies or adding the latter for greater accuracy.
To wrap up, I think more work could be performed to make this review easier to read/digest.
We thank the reviewer for these comments. We have checked and corrected the English used in the manuscript.
###########################################################################
Other comments
The manuscript needs more proof-reading, for example
in the title: Franesoid
Page 1: farnsoid
Page 2: al-so
Page 4 : PPAR-alfa
Page 7 : literuature
We thank the reviewer for these comments. We have checked and corrected typos throughout the manuscript.
Round 2
Reviewer 2 Report
I have no objection.